# Research on the Open-Categorical Classification of the Internet-of-Things Based on Generative Adversarial Networks

**Caidan Zhao \* , Mingxian Shi, Zhibiao Cai and Caiyun Chen**

Department of Communication Engineering, Xiamen University, Xiamen 361005, China;
23320171153172@stu.xmu.edu.cn (M.S.); 23320151154069@stu.xmu.edu.cn (Z.C.); chency@stu.xmu.edu.cn (C.C.)
\* Correspondence: zcd@xmu.edu.cn; Tel.: +86-592-258-0078

**Abstract:** Nowadays, it is more and more important to deal with the potential security issues of internet-of-things (IoT). Indeed, using the physical layer features of IoT wireless signals to achieve individual identity authentication is an effective way to enhance the security of IoT. However, traditional classifiers need to know all the categories in advance to get the recognition models. Realistically, it is difficult to collect all types of samples, which will result in some mistakes that the unknown target class may be decided as a known one. Consequently, this paper constructs an improving open-categorical classification model based on the generative adversarial networks (OCC-GAN) to solve the above problems. Here, we have modified the loss function of the generative model G and the discriminative model D. Compared to the traditional GAN model which can generate the fake sample overlapping with the real samples, our proposed G model generates the fake samples as negative samples which are evenly surrounding with the real samples, while the D model learns to distinguish between real samples and fake samples. Besides, we add auxiliary training not only to gain a better recognition result but also to improve the efficiency of the model. Furthermore, Our proposed model is verified through experimental study. Compared to other common methods, such as one-class support vector machine (OC-SVM) and one-versus-rest support vector machine (OvR-SVM), the OCC-GAN model has a better performance. The recognition rate of the OCC-GAN model can reach more than 90% with a recall rate of 97% by the data of the IoT module.

**Keywords:** internet-of-things; physical layer security; open-categorical classification; generative adversarial networks

## 1. Introduction

With the study advances on IoT technologies, the IoT devices have a wide application in our daily life, e.g., a smart city in which the IoT devices belonging to different application contexts cooperate for providing services related to e-health, smart factories, energy and traffic management [1,2]. According to Gartner, IoT product and service suppliers will generate incremental revenue exceeding $300 billion, mostly in services, in 2020 [3]. Additionally, in the emerging domain of the IoT, cyberphysical IoT devices has become a novel type of digital resources, which is a physical object augmented with sensing/actuation, processing, storing, and networking capabilities [4]. All these devices provide a set of physical and digital services to humans, machines, or digital systems. However, the development of IoT devices brings not only lots of conveniences but also enormous problems for the security. Several papers (e.g., [5–9]) have studied related security issues. And a recent study by Hewlett Packard revealed that 70% of the most commonly used IoT devices contained serious vulnerabilities [10]. Once they have been attacked, many of the sensitive data (privacy for medical data, personal routine data, etc.) may be revealed to unknown sources, which will result in identity

theft. For example, in 2016, the hackers generated a large number of DDoS attacks on the Dynamic Domain Name System (Dyn-DNS) server through the vulnerability of IoT devices, which led to the closure of many websites including the Twitter giant company [11]. As we can see, it is crucial to solve the security problems of the IoT.

Nowadays, the IoT security protection still adopts the wireless network security strategy, which determines the identity of the legitimate users by the Internet Protocol (IP) address [12,13], Media Access Control (MAC) address [14] or the key encryption scheme [15]. To be specific, In the Open System Interconnection (OSI) model [16], the third layer of the network layer is responsible for the IP address, and the second layer of the data link layer is responsible for the MAC address. The MAC address is used to uniquely identify a network card in the network. If one device has one or more network cards, each network card needs and has a unique MAC address. Because the MAC address consists of hexadecimal digits, the intruders can illegally attack IoT devices by counterfeiting the above identifications. So, we consider the features of the wireless signals as the individual identification feature of IoT devices, which can accurately identify unauthorized users who are attempting to join the network. For example, not only the DDoS attack but also the other attacks will be prevented in the physical layer. In brief, physical layer security is an effective way to enhance the security of the IoT.

In general, the process of detection and recognition of the wireless signals is composed of collection, extraction, training, and classification. First of all, we need to collect wireless signals and extract the features representing the individual identification. Then, it is important to obtain the trained classification by training the classifier adaptively. Finally, the model of the classifier can be used to detect the identification of the IoT devices. As for the classification, the algorithms of machine learning include k-nearest neighbor (KNN) classifier [17], back propagation (BP) neural network classifier [18] and support vector machine (SVM) classifier [19], etc., but the above algorithms are all used for closed-set recognition, which is defined that the test sets are all known classes. Machine learning models trained based on closed-set data can only identify the samples that are the same prior probability distributions with the training sets. For unknown classes, the model only discriminates as a known class in the closed set with errors. On the one hand, in the real world, because we are unable to collect all types of samples, the problems of open-categorical classification (OCC) draw more and more attention recently. On the other hand, high integration of IoT devices has very little difference in individual characteristics, which is more difficult to achieve accurate results for OCC. Therefore, this paper proposes an IoT OCC algorithm based on generative adversarial networks (GAN) and deep learning networks. Additionally, with the measured data of the collected IoT module, results of experimental applications of this model are given to illustrate the proposed technique. The contributions of this paper are as follows:

- We propose an OCC algorithm based on improving GAN.
- According to the theoretical basis of OCC, this paper analyzes the feature distribution generated by GAN and the performance of the trained discriminator.
- We use the measured IoT data to conduct experiments, and compare the proposed algorithm with other related work about OCC.

In this paper, we modify the loss functions of the generative model and the discriminative model. On the other hand, we also add auxiliary training to improve the efficiency of the OCC-GAN model. In short, the result of the experimental application of this OCC-GAN model analysis procedure are given to illustrate the proposed technique. We collect the IoT module ($E05$-$MLE124AP2$) to construct the experiment, and the results indicate that with the increasing of the test classes, the OCC rate decreases slowly, but the recognition rate of overall unknown class is above 90%. Additionally, the recall rate of the known classes reaches 97%. Besides, the other common methods are compared to the OCC-GAN model, which proved that our proposed model had a better performance.

The rest of this paper is organized as follows. Section 2 explains the use of physical layer security, and surveys the previous research related to OCC and GAN. Section 3 is developed to point out the

structure of the basic GAN model and provides the framework of the generator and the discriminator model. Section 4 describes the model of OCC-GAN itself in a general way, and also discusses how to evaluate system performance. The results of experiments are interpreted in Section 5. Section 6 contains some conclusions and some ideas for further work.

## 2. Related Work

The related work includes three parts. Section 2.1 briefly describes the physical layer security, while Section 2.2 looks at the knowledge of open-categorical classification, and the GAN are considered in Section 2.3.

### 2.1. Physical Layer Security

Physical layer security refers to the use of the characteristics of physical layer wireless signals to protect the security of computer equipment and data. It is the lowest barrier of wireless network security, which can accurately identify authorized and unauthorized users. In the process of research on individual identification, many scholars have proved that it is an effective method of identity authentication based on characteristics of the wireless signals [20–22]. Since the signals transmitted by each device imply its individual characteristics, such as signal envelope fluctuations, steepness, modulated instantaneous frequencies, and wireless transmission paths, these physical layer features are unique and unclonable. Thus, we can use wireless signals to identify "legal" or "illegal" IoT devices, thereby ensuring IoT security.

### 2.2. Open-Categorical Classification

There are two situations in the recognition system: closed-set and open-set identification. Closed-set recognition refers to the fact that the test set is a known category. On the contrary, the open-set recognition is that the test sets have the unknown classes. In the theory of statistical learning, it is often only focused on the recognition of closed sets. However, in reality, open-categorical detection has a great practical value and application prospect, such as ignorant obstacle recognition in automatic driving [23], open-categorical aircraft recognition in image processing [24], stranger detection in speech recognition [25], open-categorical text classification [26,27], open-categorical recognition of wireless signals [28], open-categorical identification of IoT wireless sensors [29], voice signal authorization of mobile phones [30], etc.

OCC has both similarities and dissimilarities with outlier detection, anomaly detection, and novelty target recognition. In the model of outlier detection, the purpose of training the system is to detect equipment trouble caused by mechanical faults, changes in system behavior, human errors, or instrument errors, etc. Blouvshtein, L. et al. [31] propose a method based on a triangular geometric algorithm to detect and filter out the abnormal points. Moreover, Sayyed, S. et al. [32] define that extraction of unknown and new data from the huge dataset that has been left during the classification process is considered as novelty detection. In comparison to the above definitions, the problem is defined as OCC if the training classes are only partly known beforehand [33].

Recently, the techniques of OCC have made great progress. One-class Support Vector Machine (OC-SVM) aims to detect outliers relative to a single training class [34], and Sheirer et al. propose "1-vs-set Machine" for single-class detection and Weibull-calibrated SVM (W-SVM) for multi-class classification in open set scenarios [35,36], so that, the one-versus-rest multi-label support vector machine (OvR-SVM) is one of the most popular algorithms for OCC. Besides, Mao C. et al. propose distributed networks for open set learning that can learn and model different novel classes [37]. Unfortunately, although the above methods for OCC have achieved some good results, there are still difficulties in theoretical support and practical applications. In the individual identification of IoT devices, because of the high similarity of each device, the mapped feature spaces are often very close. To have a better performance for open-categorical individual identification of the same batch of IoT devices, further improvement of the model construction is required.

### 2.3. Generative Adversarial Networks

Nowadays, the wide applications of GAN have made a series of achievements. Chen L. et al. achieve cross-model audio-visual of musical performances based on conditional GAN [38]. A model to predict the next frame in the video sequence is trained with the adversarial loss, by Lotter W. et al. [39], and Ledig C. et al. present a GAN for image super-resolution [40]. Additionally, for audio signal, a sequence generation framework, called SeqGAN, is proposed by Yu L. et al. [41]. In the future, there is an increasing need for expanding the application of GAN.

In fact, the basic principle of GAN is to play a mutual game between the generative model G and the discriminative model D. Here, a G model captures the data distribution, and a D model estimates the probability that a sample comes from the training data rather than from G [42]. The main difficulty of the OCC problems is the lack of negative samples to train the identification model. On the basis of the concept mentioned above, we are able to consider that the G model can generate fake samples as the negative samples. When we run a generative model, $G = G(z)$ receives random noise $z$ as input. After an optimal training procedure, the distribution of $G(z)$ is shown to match the distribution of the real samples, which can be regarded as the fake samples [43]. To deal with the OCC problems, the ideal distribution of the fake sample may be evenly distributed in the open space. Even if the sample data are small, the generated fake sample can be trained as the negative sample to gain the discriminative feature space. So, it is obvious that GAN model can deal with the challenges of OCC.

## 3. Network Overview

### 3.1. GAN Model

The GAN model includes a generative model G and a discriminative model D, and the composition of G and D can be a convolutional neural network (CNN) or a Multi-Layer Perceptron (MLP) network, as shown in Figure 1.

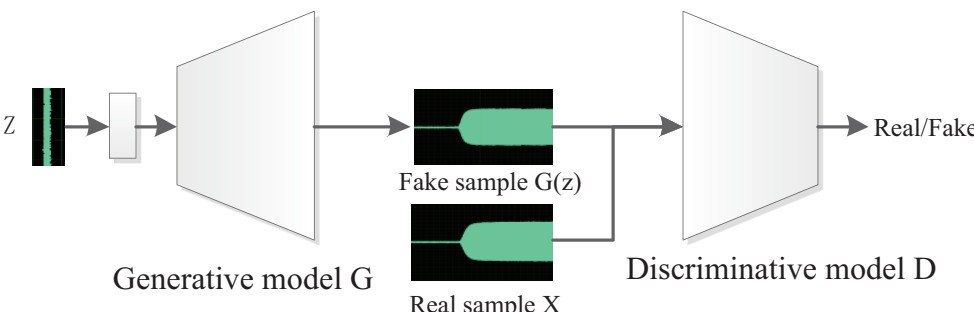

**Figure 1.** GAN model.

The generative model G gets a generated sample $X_{fake} = G(z)$ by inputting a random signal $z$ that match a certain distribution, or an additional control variable $c$. The discriminative model D obtains a probability value by distinguishing a generated sample from a real one, and the loss function of the G and D model are as follows:

$$L_D = E_z[log(1 - D(X_{fake}))] + E_X[logD(X_{real})] \tag{1}$$

$$L_G = E_z[log(1 - D(X_{fake}))] \tag{2}$$

Here, D is trained to maximize $L_D$ while G is trained to minimize $L_G$. In the physical sense, the Equation (1) is that the D model learns to distinguish between real samples and fake samples, and Equation (2) aims that a generated fake sample by the G model is more and more difficult to distinguish. At last, these two models reach a balance.

In fact, the GAN may often occur mode collapse, which means the generative samples does not match with the characteristics of the training set, even far away. In order to achieve the balance of G and D, the loss function for the two models should keep balance until stable. Goodfellow, I. et al. put forward the algorithm that the loss is based on the ordinary distance measurement of Kullback–Leibler (KL) divergence and Jensen–Shanno (JS) divergence [42]. Yet, this algorithm has some defects in the GAN application. Firstly, there are different loss degrees for the two models in training, which results in unbalance of them. Secondly, the feature space of the training samples and the generated samples may not coincide, so that the loss of the generator is close to zero, while the discriminator model can accurately identify the real and fake samples each time. All lead that the system cannot continue to train. In [44], the engineering method is used to improve the GAN. At the same time, the Wasserstein GAN (WGAN) algorithm is proposed in [45]. the Wasserstein distance takes the place of KL divergence, which makes models basically stable. Considering that the ultimate goal is to generate non-coincident models as much as possible, the WGAN algorithm still has a large loss when it is not coincident. In short, the OCC model of this paper is based on the Deep Convolutional GAN (DCGAN) [44].

*3.2. Wireless Signals GAN Model*

So as to achieve better balance, the D model must match the performance of G model in the GAN. The discriminative model D plays two roles in the GAN. One is that can output a predicted value comparing with the real label. Another is that the intermediate feature layer of the discriminator D can be used as unsupervised feature extraction. The structure of the model is shown in Figure 2.

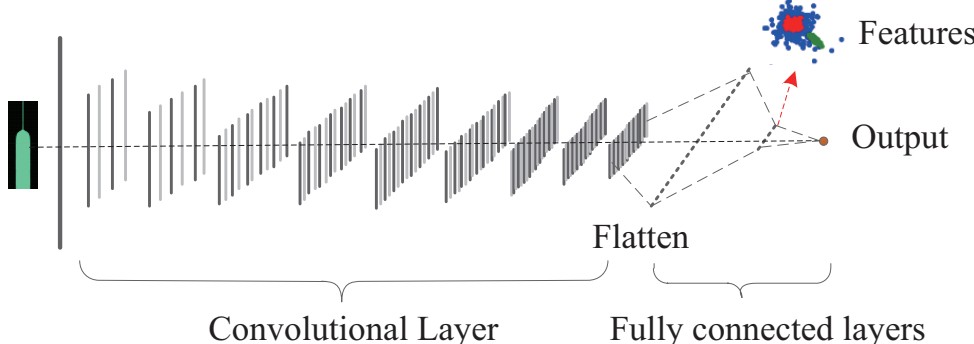

**Figure 2.** Framework of the discriminator D in the GAN.

the D model of the GAN is designed as the deep convolutional neural network, as shown in Figure 2. Here, the dimension of the input sample signal feature reaches 4096. The numbers of feature channels of each convolutional layer are $[16, 32, 32, 64, 64, 128, 128, 256, 256]$, and the corresponding feature dimensions are $[2048, 1024, 512, 256, 128, 64, 32, 16, 8]$. Especially, the convolution kernel of each layer is $31 \times 1$, whose nonlinearly mapped function is based on the *Leakyrelu*. Meanwhile, the stride is 2. At last, the output data of the final convolutional layer transpose into one-dimensional features after through the fully connected layers, whose dimension is $1 \times 8$. At the moment, the output of the penultimate layer can be used as the feature, while the output of the last dimension is to judge whether the input signal is real or fake.

To keep the balance of D and G model, the framework of the generator G is similar to the discriminator D without the fully connected layers, but the convolutional layers of the G model is defined as deconvolution, whose numbers of channels are $[256, 256, 128, 128, 64, 64, 32, 32, 16, 1]$, and the corresponding feature dimensions are $[8, 16, 32, 64, 128, 256, 512, 1024, 2048, 4096]$. The convolution kernel of each layer is also 31. The middle layer takes *Prelu* as the nonlinear mapping, while the last layer takes *tanh* as the activation function.

It is important to choose the correct loss function to achieve the balance of the GAN model. In this paper, we utilize the mean square error to calculate the degree of loss, as Equation (3).

Also, a regularized loss term is added to the generator at the beginning of the training as in Equation (4) to achieve stable training quickly.

$$D = avg((h(x) - label)^2)$$ (3)

$$L_2 = avg(||x_{real} - x_{fake}||)$$ (4)

## 4. Open-Categorical Classification Based On GAN

It is difficult to collect all possible sample classes in reality; here will be some unmarked categories, called unknown classes. The machine learning algorithm based on the statistical learning method cannot use the prior probability knowledge to identify an unknown class. Therefore, it is important to construct a more accurate model to deal with the unknown classes. In order to solve this problem, we propose the OCC model based on the generative adversarial networks (OCC-GAN), as shown in Figure 3.

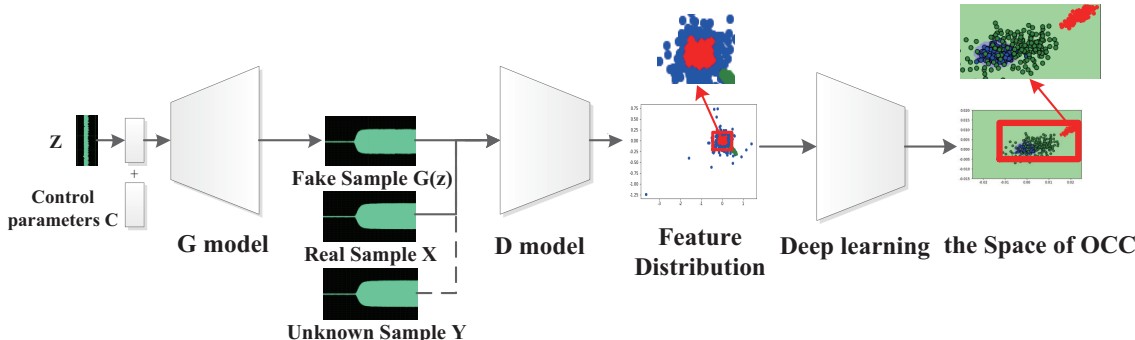

**Figure 3.** Framework of the OCC-GAN.

Many researchers have noticed this OCC problem and proposed many excellent solutions [35,36,46]. However, there are still some shortcomings while we make use of the similarity or the distance threshold as the inspection standard. All these methods require that the feature distribution space is all included in the hypersphere, but this is not the case in reality. Suppose the feature distribution of the sample is the shape of Figure 4 [47]; under this circumstance, there are errors when we take advantage of the distance threshold.

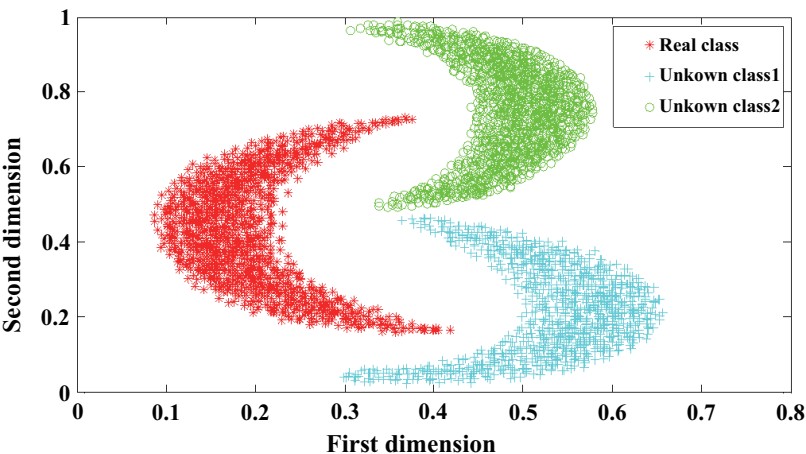

**Figure 4.** Feature distribution.

Meanwhile, many researchers propose the Equation (5), which is used to measure the risk of the OCC model. The $f(x)$ is the measurable space, $O$ is the model discriminant area after training, and $S$

is the measurable space of the entire open set. If the discriminant region of the model can approach the boundary of the feature distribution of the known class, the risk of the model will be lower.

$$R_O = \frac{\int_O f(x)dx}{\int_S f(x)dx}$$

(5)

According to Equation (5), suppose some negative samples can be obtained, whose feature space can tightly surround the feature distribution space of the known class. As shown in Figure 5, the red known samples are surrounded by the blue samples.

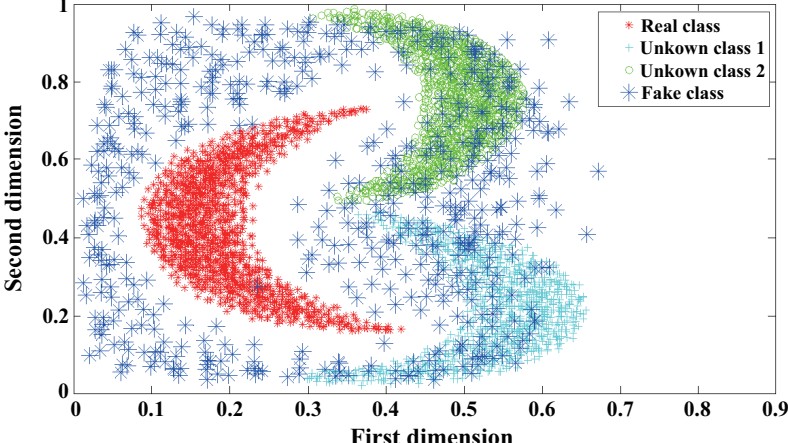

**Figure 5.** Feature distribution of the generated fake samples.

Based on the feature distribution in Figure 5, it is easy to obtain a discriminative space by SVM or Deep Neural Networks (DNN) algorithms, so that the classifier can discriminate the sample in the space into a known class. Otherwise, it is an unknown one. The goal of the OCC is to find a distribution space similar to the surrounding gray area in the Figure 6 that the discriminative space is closer to the distribution of the known class.

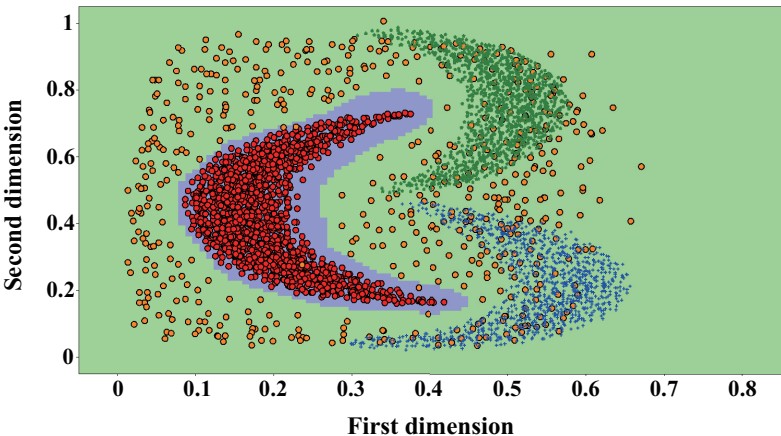

**Figure 6.** Discriminating space based on the SVM model.

In fact, since the size of the open space is completely unknown, the negative samples as described above cannot be obtained. To solve this problem, we can generate the fake samples based on GAN surrounding with the known classes. However, it is difficult to generate the actual feature distribution as we want. As shown in Figure 7, the red points are the real samples and the blue points represent

the fake samples. It is obvious that the two types of feature distributions are overlapping rather than surrounding. Therefore, it is difficult to directly take samples generated by the GAN as negative classes.

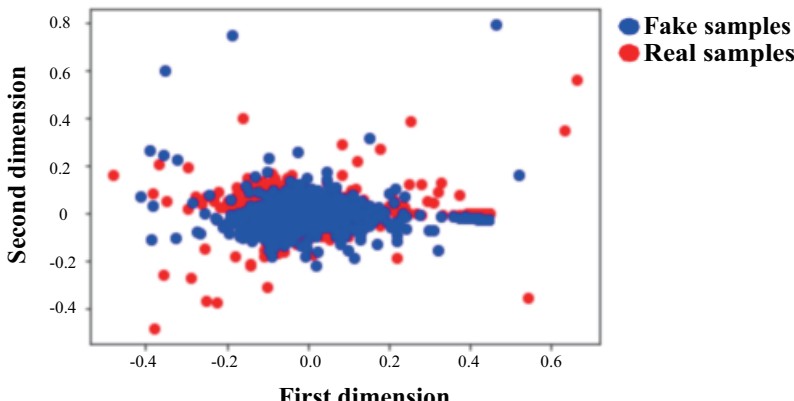

**Figure 7.** Feature distribution by the GAN without modification.

Thus, the solution lies in the improvement of the GAN model to generate the desired samples, as shown in Equations (6)–(14).

The loss function of discriminative model D is:

$$P_D(\hat{x}) = avg((H(\hat{x}) - label_{fake})^2) \tag{6}$$

$$P_D(x) = avg((H(x) - label_{real})^2) \tag{7}$$

$$L_D^k = \arg\min_D P_D(\hat{x}) + P_D(x) \tag{8}$$

The loss function of generative model G is:

$$L_1(\hat{x}, X_k) = \max\{0, \arg\min_{x \in X_k} dist(F(x, \hat{x})) - \mu_1\} \tag{9}$$

$$L_2(\hat{x}, X_k^-) = \max\{0, \mu_2 - \arg\min_{x \in X_k^-} dist(F(x, \hat{x}))\} \tag{10}$$

$$L_3(\hat{x}, X_k) = \max\{0, \mu_3 - \arg\min_{x \in X_k} dist(F(x, \hat{x}))\} \tag{11}$$

$$J = avg(||x_{real} - x_{fake}||^2) \tag{12}$$

$$P_D(\hat{x}) = avg((H(\hat{x}) - label_{real})^2) \tag{13}$$

$$L_G^k = \arg\min_G P_D(\hat{x}) + \lambda_0 J + \lambda_1 L_1(\hat{x}, X_k) + \lambda_2 L_2(\hat{x}, X_k^-) + \lambda_3 L_3(\hat{x}, X_k) \tag{14}$$

where $x$ is the real training sample, $\hat{x}$ is the fake sample. $X_k$ is the space of the $k$th real class. Similarly, $X_k^-$ means the space of the $k$th fake class. $\mu_1$ and $\mu_2$ is the average distance between the sample features of $X_k$ or $X_k^-$, while $u_3$ is the maximum distance of the $X_k$ samples and plus the standard deviation. Here, $H(\bullet)$ indicates the decision output of the GAN model, $F(\bullet)$ indicates the features output of the GAN model. $dist(\bullet)$ is the Euclidean distance. In brief, $L_1$ represents the loss of the fake sample in the space of $X_k$, $L_2$ represents the loss that the fake sample is to close to the previous fake sample. $L_3$ represents the loss that a fake sample point is too far away from the space of $X_k$. $J$ is regularization, whose value is the euclidean distance between the real training sample and the generated fake sample, and $P_D(\bullet)$ is the loss term, $\lambda$ is a hyperparameter.

Based on the above theory, we train several real IoT device wireless signals as shown in Figure 8. The gray part shows the discriminative area of the SVM, blue points are the real training samples,

and green points are the generated fake samples. It can be seen that the training sample is surrounded by the generated sample. If the feature distribution of the unknown class is far from the real sample, it will be mapped outside the gray area. According to the probability of the SVM, we can learn whether the test sample class belongs to the unknown one. Besides, it can give the probability as well.

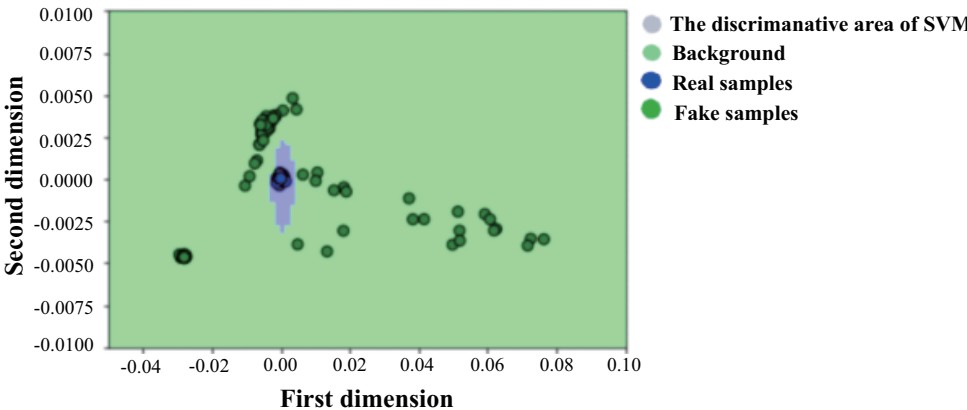

**Figure 8.** Discriminating space by the improved GAN.

However, the model sometimes is unstable during training. The fake samples generated by the GAN model do not completely surround the training samples. As shown in Figure 9, most of them tend to be in the upper right corner, and the vacancies exist in the lower left corner. As we can know, the sample is too scattered and not uniform enough to closely surround the feature distribution of known classes.

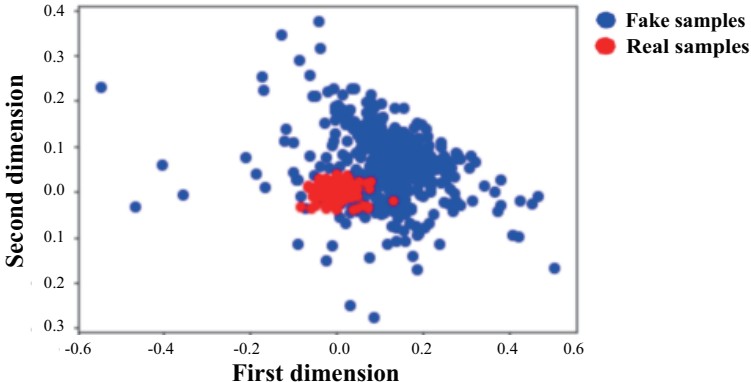

**Figure 9.** Feature distribution of the unstable GAN.

Therefore, we improved the algorithm and divided Equations (6) to (14) into two steps. The training of the GAN model is separated from the process of the generated fake samples, as shown in Equations (15) and (16). Additionally, we change the $dist(\bullet)$ to the cosine similarity distance, as shown in Equation (17).

$$L_G^k = \arg\min_G P_D(\hat{x}) + \lambda_0 J \tag{15}$$

$$L_G^k = \arg\min_G \lambda_1 L_1(\hat{x}, X_k) + \lambda_2 L_2(\hat{x}, X_k^-) + \lambda_3 L_3(\hat{x}, X_k) \tag{16}$$

$$Cossin(x, y) = \frac{\sum_i x_i y_j}{\sum_i x_i^2 \sum_i y_i^2} \tag{17}$$

The model is divided into two steps. Firstly, we can train the GAN model until stable. After that, the D model is stopped, while the G model is trained according to Equation (16). The stable result

is illustrated in Figure 10, the green points in the figure are the fake sample points generated by the model, the blue points are the known class of training, and the red points are the known class of the test. Obviously, the red and blue points are surrounded by a gray area, which is the discriminant space. It can be seen that the gray area closely surrounds the spatial distribution of the known class, and according to Equation (5), the risk is greatly reduced.

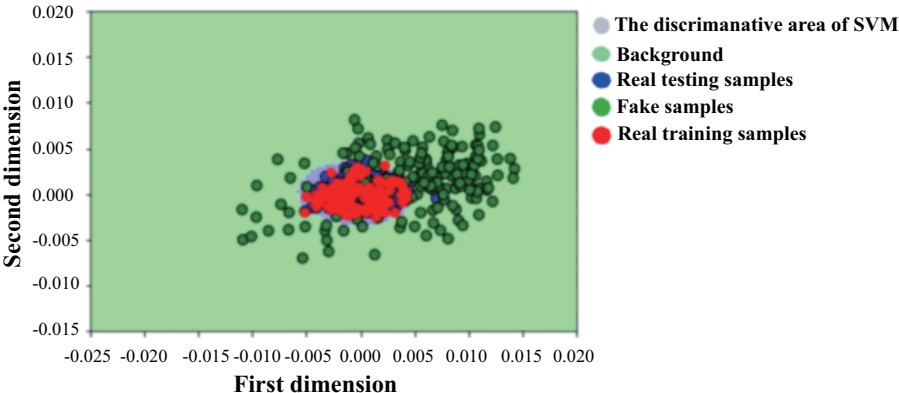

**Figure 10.** Discriminating space based on the stable GAN.

For the above model, on the one hand, the previous training was conducted in a single class. On the other hand, during the experiment, we find that not all classes have a good result. Sometime, the feature space of the unknown class may map to the area which overlaps slightly with the known class. So we add auxiliary training to the previous single-class OCC model, which enabled the OCC-GAN model to update effectively, as shown in Equations (18) and (19). Since the lost function of the D model is changed, we make the G model training 3 times while the D model training once to achieve the balance of training against the network.

$$P_D(x_a) = avg((H(x_a) - label_{\text{auxiliary}})^2) \tag{18}$$

$$L_D^k = \arg\min_D P_D(\hat{x}) + P_D(x) + P_D(x_a) \tag{19}$$

There are five IoT devices in the test, and all the wireless signals of each device are marked. Suppose the first to fourth classes are known and the fifth class is unknown. When the first class is trained in the model, the second class to the fourth class is added as the auxiliary class. The effect after training is shown in Figure 11, the red points are the first category, the blue points are the generated fake samples, and the green points are the fifth unknown devices.

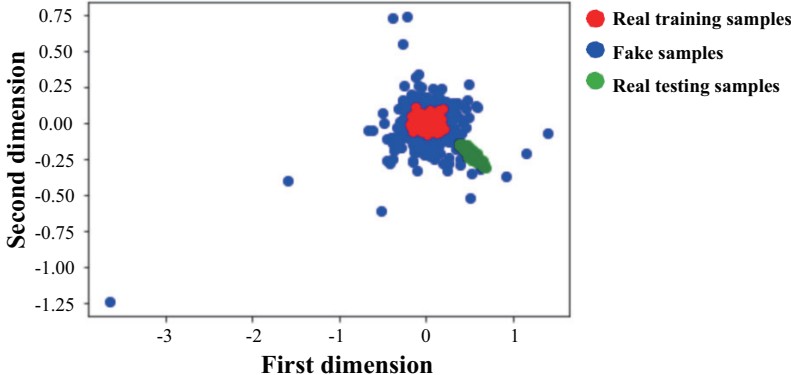

**Figure 11.** Feature distribution based on the auxiliary classes GAN.

## 5. Experimental Result

As shown in Figure 12, the experimental data are acquired by universal software radio peripheral (USRP) with a sampling frequency of 48 MHz and a bandwidth of 10 MHz, and the type of IoT module is $E05\text{-}MLE124AP2$, which is a wireless transceiver module, operates at 2.4 GHz with small-size and 100 mW transmitting power. Because of its stable performance, small-size by professional hard design and being commonly used, this module are convenient for all kinds of embedded development. Therefore, we take this module representing the IoT devices to construct our experiment.

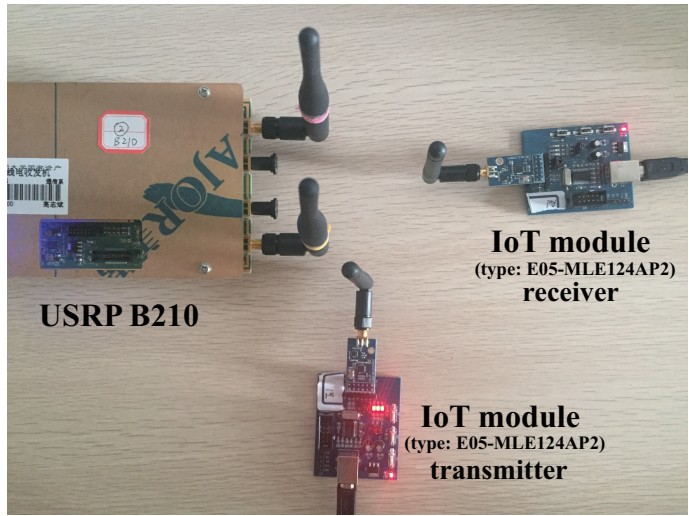

**Figure 12.** Experimental equipment.

Here, we collect the signals of the transmit module with the same receive module as the training samples, and the No.1 to No.3 IoT equipment are used as training classes, and No.5 to No.19 equipment are tested as unknown classes. Then, the test results are averaged after training. As shown in Figure 13, with the increasing of the test classes, the OCC rate decreases slowly, and the recognition rate of overall unknown class is above 90%. The OCC model not only needs to have an accurate identification of the unknown classes, but also the known classes. To this end, with No.1 to No.3 test samples inputting into this model, the recall rate of the known classes is 97.0% after average. It can be seen that the model can better identify the known classes.

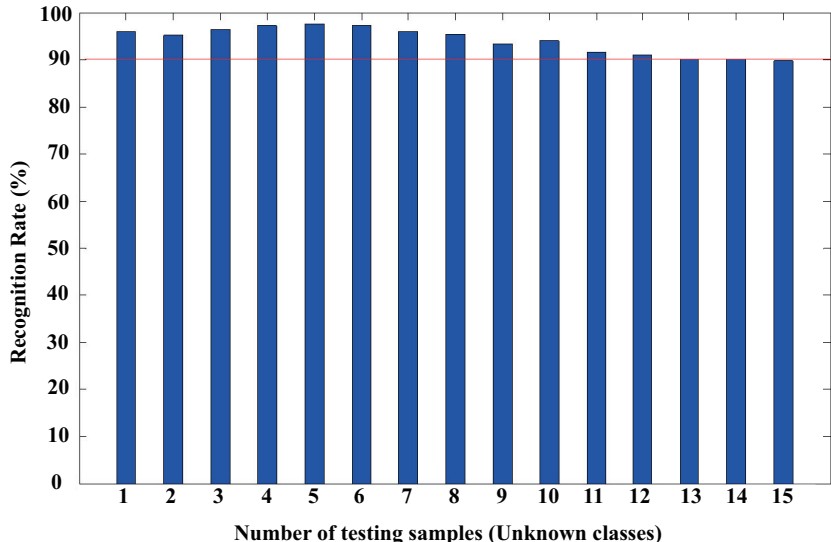

**Figure 13.** Result of the OCC-GAN model.

OvR-SVM is one of the most common approaches to deal with the multi-class OCC problems based on the "one-to-many" training method, which is often used to construct multi-class identification. The improving OvR-SVM algorithm can get both class labels and probability values, while the OvR-SVM model detects the sample as an unknown class by setting a probability threshold. If the probability of the output is less than the threshold, the model will determine that the test sample is an unknown class. Otherwise, it is determined to be a known class. Figure 14 shows the average performance of the model for the test data. We can find that the recall rate of the model is high when the threshold is large, but the rate of the unknown classes is very low. While the threshold is small, the accuracy of the model increases rapidly, and the recall rate decreases rapidly. It can be seen from the horizontal axis that when the threshold is reduced from 0.600049 to 0.6, the performance of the model is seriously affected by the threshold.

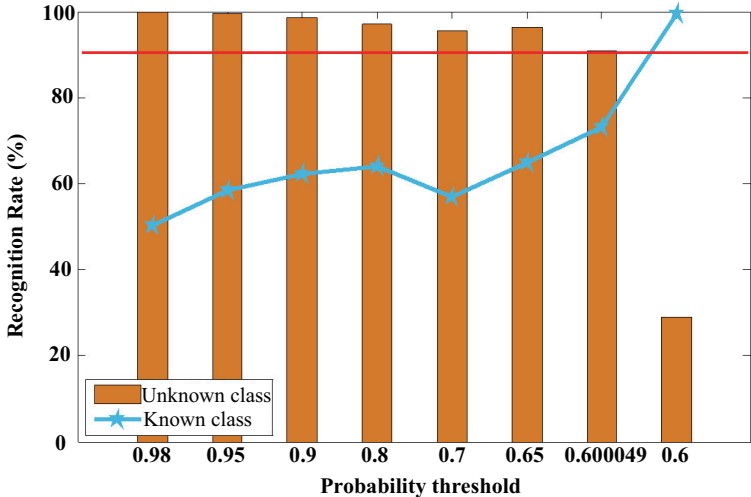

**Figure 14.** Recognition rate of the OvR-SVM algorithm for OCC.

In order to compare the OvR-SVM with the OCC-GAN model, we set the threshold of this experiment as 0.600049 and train the models with the same 15 unknown test classes. The experimental results are shown in Figure 15. When the number of the test class is small, the unknown class recognition effect of the OCC-GAN model is slightly better. On the contrary, while the number of the test samples is increasing, the unknown class recognition effect of the OvR-SVM model is better. However, the recall rate of OvR-SVM is only 73%, which is far lower than the recall rate of 97% of the OCC-GAN model.

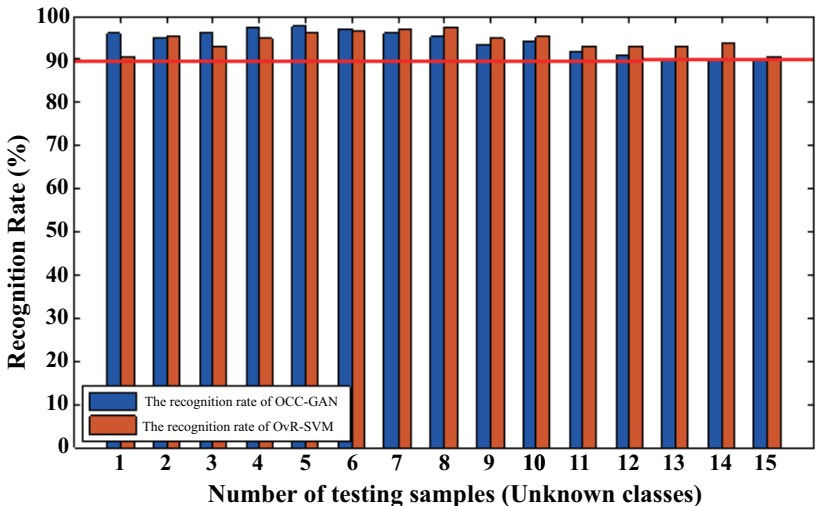

**Figure 15.** Comparison of the OvR-SVM and OCC-GAN algorithms.

In addition to the OvR-SVM algorithm, we also use the current OC-SVM algorithm for comparison. It is based on the Gaussian kernel nonlinear mapping. The features of each class are mapped to the same distribution space, and the spatial distance between the test class and the training class is used to determine whether it is an unknown category. The experimental comparison results are shown in Figure 16 and the overall performance of the OC-SVM is slightly lower than the OCC-GAN model. The recall rate of OCC-GAN is 97.0% as much as OC-SVM, and the accuracy of OC-SVM is only 88%.

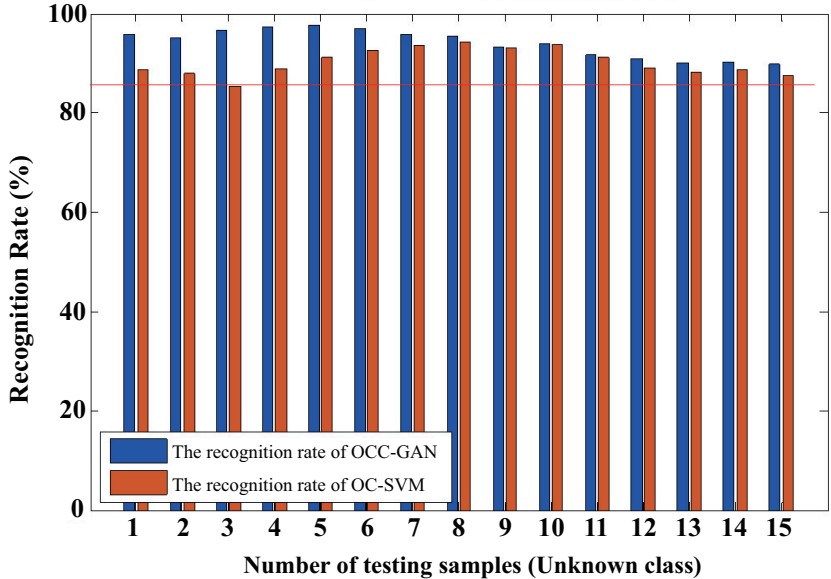

**Figure 16.** Comparison of the OC-SVM and OCC-GAN algorithms.

In comparison with the performance of the three algorithms above, the *F*1 is defined to measure, as shown in Equation (20). We can get the result in Table 1, and the performance of the OCC-GAN model is the best.

$$F1 = \frac{2 * RecognitionRate_{unknown} * RecallRate_{known}}{RecognitionRate_{unknown} + RecallRate_{known}} \tag{20}$$

**Table 1.** The result of *F*1.

|      | OvR-SVM | OC-SVM | OCC-GAN |
|------|---------|--------|---------|
| *F*1 | 0.802   | 0.923  | 0.939   |

## 6. Conclusions

OCC is an inevitable problem in individual identification. Based on the open-categorical risk theory, this paper proposed the OCC-GAN model, which generated fake samples to be evenly distributed in the open space. Moreover, an improved algorithm of incremental OCC-GAN enabled the model to continuously integrate new sample classes with a high recognition rate of OCC. The final identification model can automatically extract features and have strong robustness. Even if it dealt with micro-data, the model can still achieve good recognition results. We conducted a series of experiments to verify the performance of our proposed model. compared to other common methods, such as OC-SVM and OvR-SVM, the OCC-GAN model had a better result. This model can achieve an accuracy rate of more than 90% with a recall rate of 97%.

However, there still exits some limitations of this method. First of all, the current model is based on a single unknown class for training. It may not make full use of the multiple unknown classes to train the networks simultaneously, which will result in a waste of resources. Then, because of the limitation

of GAN, the generated fake samples are not surrounding the real samples sometimes, which will make the model unstable. Additionally, in this experiment scenario, we didn't consider the distance between the IoT module and the collection system, which ignored the noise of the environment. In the future, in order to deal with the multiple unknown class network, we hope to consider Auxiliary Classifier GAN (ACGAN) or other technologies to train the networks based on multiple classes. On the other hand, future research will focus on improving the convergence of this model and consider to add the random noise to simulate different distances.

**Author Contributions:** C.Z. presented the idea and designed the proposed algorithm; M.S. and Z.C. built the simulation model and performed the simulation; M.S. and C.C. wrote the first draft of the manuscript.

**Funding:** This research was funded by the National Natural Science Foundation of China under Grant No. 91638204.

**Acknowledgments:** The authors would like to express their acknowledgement for the support from the National Natural Science Foundation of China under Grant No. 91638204.

**Conflicts of Interest:** The authors declare no conflict of interest.

## Abbreviations

The following abbreviations are used in this paper:

| | |
|---|---|
| IoT | internet-of-things |
| DDoS | Distributed Denial of Servic |
| OCC | Open-Categorical Classification |
| GAN | Generative Adversarial Networks |
| OCC-GAN | Open-Categorical Classification based on Generative Adversarial Networks |
| SVM | Support Vector Machine |
| OC-SVM | One-Class Support Vector Machine |
| OvR-SVM | One-versus-Rest Multi-Label Support Vector Machine |
| WGAN | Wasserstein Generative Adversarial Networks |
| DCGAN | Deep Convolutional Generative Adversarial Networks |
| ACGAN | Auxiliary Classifier Generative Adversarial Networks |

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
