# Peer review of "Research on the Open-Categorical Classification of the Internet-of-Things Based on Generative Adversarial Networks"

_applsci, doi:10.3390/app8122351_

Round 1
Reviewer 1 Report
The paper addresses an interesting and timely topic, that how to prevent the Internet of things (IoT) from potential security issues. The authors developed an open-categorical classification model based on the Generative Adversarial Networks. The paper has many strengths and the model is verified through experimental study. However, I have some comments which I would like to be addressed before the acceptance of the paper.
Major Comments:
1. I recommend the authors to follow the MDPI’s proper abstract format. Define the problem statement that needs to be addressed in the researched field. Explain the important experimental findings and results and above all please explain the novelty of the propounded results in the abstract.
2. Please provide the summary of the most important results in the last paragraph of the introduction section.
3. Add more explanation in section 2.3 about the use of fake samples.
4. Why authors use IoT module E05-MLE124AP2 and what is its significance?
5. Modify the conclusion section and explain in detail the limitations of the current model.
Minor comments:
1. Page 1, line 22, 23: Please provide the reference to the claim for readers’ convenience.
2. Page 3: When starting a new sentence use the author’s name rather than just putting the reference number. Made changes accordingly into the references 25, 27,28,29,30 on page 3.
3. Page 6: line 387,388: Please provide the reference to the claim for readers’ convenience.
Author Response
Thank you for your letter and for the reviewers’ comments concerning our manuscript entitled “Research on the Open-Categorical Classification of the Internet of Things Based on the Generative Adversarial Networks” (ID: applsci-389985). Those comments are all valuable and very helpful for revising and improving our paper. The main corrections in the paper and the responds to the reviewer’s comments are shown in the attached Word file.
Point 1: I recommend the authors to follow the MDPI’s proper abstract format. Define the problem statement that needs to be addressed in the researched field. Explain the important experimental findings and results and above all please explain the novelty of the propounded results in the abstract.
Response 1: We have re-written this part according to the reviewer’s suggestion. Here, we add the novelty of the model, and give important experimental results in detail. And we have made the text bold and placed an underline for the revised part of the abstract. Our abstract is shown as follows:
Nowadays, it is more and more important to deal with the potential security issues of Internet of Things (IoT). Indeed, using the physical layer features of IoT wireless signals to achieve individual identity authentication is an effective way to enhance the security of IoT. However, traditional classifiers need to know all the categories in advance to get the recognition models. Realistically, it is difficult to collect all types of samples, which will result in some mistakes that the unknown target class may be decided as a known one. Consequently, this paper constructs an improving open-categorical classification model based on the Generative Adversarial Networks (OCC-GAN) to solve the above problems. Here, we have modified the loss function of the generative model G and the discriminative model D. Comparing to the traditional GAN model which can generate the fake sample overlapping with the real samples, our proposed G model generates the fake samples as negative samples which are evenly surrounding with the real samples, while the D model learns to distinguish between real samples and fake samples. Besides, we add auxiliary training not only to gain a better recognition result but also to improve the efficiency of the model. Furthermore, Our proposed model is verified through experimental study. Comparing to other common methods, such as one-class support vector machine (OC-SVM) and one-versus-rest support vector machine (OvR-SVM), the OCC-GAN model has a better performance. The recognition rate of the OCC-GAN model can reach more than 90% with a recall rate of 97% by the data of the IoT module.
Point 2: Please provide the summary of the most important results in the last paragraph of the introduction section.
Response 2: We have added the summary in the introduction section. We describe this part as follows:
In this paper, we modify the loss functions of the generative model and the discriminative model. Meanwhile, we also add auxiliary training to improve the efficiency of the OCC-GAN model. In short, the result of the experimental application of this OCC-GAN model analysis procedure is given to illustrate the proposed technique. We collect the IoT module (E05 − MLE124AP2) to construct the experiment. And the results indicate that with the increasing of the test classes, the OCC rate decreases slowly. But the recognition rate of overall unknown class is above 90%. Additionally, the recall rate of the known classes reaches 97%. Besides, the other common methods are compared to the OCC-GAN model, which proved that our proposed model had a better performance.
Point 3: Add more explanation in section 2.3 about the use of fake samples.
Response 3: As the reviewer suggested that provide more explanation about the use of fake samples, shown as follows:
The principle of GAN is to play a mutual game between the generative model G and the discriminative model D. Here, a G model captures the data distribution, and a D model estimates the probability that a sample comes from the training data rather than from G. the main difficulty of the OCC problems is the lack of negative samples to train the identification model. On the basis of the concept mentioned above, we are able to consider that the G model can generate fake samples as the negative samples. When we run a generative model, G = G(z) receives random noise z as input. After an optimal training procedure, the distribution of G(z) is shown to match the distribution of the real samples, which can be regarded as the fake samples [38]. To deal with the OCC problems, the ideal distribution of the fake sample may be evenly distributed in the open space. Even if the sample data are small, the generated fake sample can be trained as the negative sample to gain the discriminative feature space. So, it is obvious that GAN model can deal with the challenges of OCC.
[38] Gauthier, J. Conditional generative adversarial nets for convolutional face generation. Class Project for Stanford CS231N: Convolutional Neural Networks for Visual Recognition, Winter semester 2014, 2014, 2.
Point 4: Why authors use IoT module E05-MLE124AP2 and what is its significance?
Response 4: In fact, the type of IoT module is E05-MLE124AP2, which is a wireless transceiver module, operates at 2.4GHz with small-size and 100mW transmitting power. Because of its stable performance, small-size by professional hard design and commonly used, this module are convenient for all kinds of embedded development. Therefore, we take this module representing the IoT devices to construct our experiment. The experiment scenario is shown as follows.
Point 5: Modify the conclusion section and explain in detail the limitations of the current model.
Response 5: Considering the reviewer’s suggestion, we have re-written the conclusion section and explain the limitations of the current model, shown as follows:
OCC is an inevitable problem in individual identification. Based on the open-categorical risk theory, this paper proposed the OCC-GAN model, which generated the fake samples to be evenly distributed in the open space. Moreover, an improved algorithm of incremental OCC-GAN enabled the model to continuously integrate new sample classes with a high recognition rate of OCC. The final identification model can automatically extract features and have strong robustness. Even if it dealt with the micro-data, the model can still achieve good recognition results. We conducted a series of experiments to verify the performance of our proposed model. Comparing to other common methods, such as OC-SVM and OvR-SVM, the OCC-GAN model had a better result . This model can achieve an accuracy rate of more than 90% with a recall rate of 97%.
However, there still exits some limitations of this method. First of all, the current model is based on a single unknown class for training. It may not make full use of the multiple unknown classes to train the networks simultaneously, which will result in a waste of resources. Then, because of the limitation of GAN, the generated fake samples are not surrounding with the real samples sometimes, which will make the model unstable. Additionally, in this experiment scenario, we didn’t consider the distance between the IoT module and the collection system, which ignored the noise of the environment. In the future, in order to deal with the multiple unknown class network, we hope to consider Auxiliary Classifier GAN (ACGAN) or other technologies to train the networks based on multiple classes. On the other hand, future research will focus on improving the convergence of this model and consider to add the random noise to simulate different distances.
Minor comments:
Point 1: Page 1, line 22, 23: Please provide the reference to the claim for readers’ convenience.
Response 1: We are very sorry for our negligence of the reference. The related reference has been added to our manuscript. “A recent study by Hewlett Packard revealed that 70\% of the most commonly used IoT devices contained serious vulnerabilities [5]”
[5] Kristi Rawlinson, H. HP News.Website, Jul. 2014. http://www8.hp.com/us/en/hp-news/press-327release.html?id=1744676#.W8RtW9UzaUl.
Point 2: Page 3: When starting a new sentence use the author’s name rather than just putting the reference number. Made changes accordingly into the references 25, 27,28,29,30 on page 3.
Response 2: As Reviewer suggested that when starting a new sentence use the author’s name rather than the reference number. We have changed the expression of this part.
“Besides, Mao C et al. propose distributed networks for open set learning that can learn and model different novel classes [25].”
“Chen L et al. achieve cross-model audio-visual of musical performances based on conditional GAN [27]. A model to predict the next frame in the video sequence is trained with the adversarial loss, by Lotter et al. [28]. And Ledig et al. present a GAN for image superresolution [29]. Additionally, For an audio signal, a sequence generation framework, called SeGAN, is proposed by Yu L. et al. [30].”
Here, the reference number is the same with the previous manuscript. Besides, we also check all the manuscript to avoid this errors.
Point 3: Page 6: line 387,388: Please provide the reference to the claim for readers’ convenience.
Response 3: On page 6, we didn’t find the line 387,388. According to the reviewer’ expression, we think that maybe the line 187, 188. So we provide the reference to the claim for readers’ convenience.
Scheirer, W.J.; de Rezende Rocha, A.; Sapkota, A.; Boult, T.E. Toward open set recognition. IEEE transactions on pattern analysis and machine intelligence Jul. 2013, 35, 1757–1772.
Scheirer, W.J.; Jain, L.P.; Boult, T.E. Probability models for open set recognition. IEEE transactions on pattern analysis and machine intelligence Nov.2014, 36, 2317–2324.
Jain, L.P.; Scheirer, W.J.; Boult, T.E. Multi-class open set recognition using probability of inclusion. European Conference on Computer Vision, Zurich, Switzerland, Sep. 2014, pp. 393–409.

Reviewer 2 Report
The proposed manuscript proposes an interesting approach for security improvement at physical layer through a machine learning-based paradigm. The manuscript contains some typos that have to be verified, such as the following (some typos on the overall identified):
authors should use an official, academic, e-mail address, instead of a generic one;
references should be verified and need to be written as "... [X]", with a blank before the reference number;
references for KNN, BP and SVM are needed;
"rather than G" -> "rather than from G";
subsection title should start with an uppercase;
figures captions have to be concluded with a dot mark;
"GAN(DCGAN)" -> "GAN (DCGAN)"
why "x_fack" instead of "x_fake"?
verify the caption of each figure, avoiding the title inside each plot.
Author Response
Dear Editors and Reviewer :
Thank you for your letter and for the reviewers’ comments concerning our manuscript entitled “Research on the Open-Categorical Classification of the Internet of Things Based on the Generative Adversarial Networks” (ID: applsci-389985). Those comments are all valuable and very helpful for revising and improving our paper. The main corrections in the paper and the responds to the reviewer’s comments are as following:
Point 1: authors should use an official, academic, e-mail address, instead of a generic one
Response 1: We have changed our e-mail address as follows:
zcd@xmu.edu.cn (C.Z.); 23320171153172@stu.xmu.edu.cn (M.S.); 23320151154069@stu.xmu.edu.cn (Z.C.); chency@stu.xmu.edu.cn (C.C.)
Point 2: references should be verified and need to be written as "... [X]", with a blank before the reference number;
Response 2: We are very sorry for our negligence of this error. In this time, we have checked all this manuscript to avoid this problem.
Point 3: references for KNN, BP and SVM are needed;
Response 3: We have added the correspondent references for KNN, BP and SVM:
KNN:Peterson, L.E. K-nearest neighbor. Scholarpedia 2009, 4, 1883.
BP: Rumelhart, D.E.; Hinton, G.E.; Williams, R.J. Learning representations by back-propagating errors. nature 1986, 323, 533.
SVM: Hearst, M.A.; Dumais, S.T.; Osuna, E.; Platt, J.; Scholkopf, B. Support vector machines. IEEE Intelligent Systems and their applications 1998, 13, 18–28.
Point 4: "rather than G" -> "rather than from G";
Response 4: In previous manuscript, line 111-112 in 3 page, the statements of “and a D model estimates the probability that a sample comes from the training data rather than G” were corrected as “and a D model estimates the probability that a sample comes from the training data rather than from G” .
Point 5: figures captions have to be concluded with a dot mark;
Response 5: We have made correction according to the reviewer’s comments.
Point 6: "GAN(DCGAN)" -> "GAN (DCGAN)"
Response 6: We are very sorry for our carelessness. The blank before the brackets was added.
Point 7: why "x_fack" instead of "x_fake"?
Response 7: We are very sorry for our incorrect writing. We have gone through all the manuscript to remove these errors.
Point 8: verify the caption of each figure, avoiding the title inside each plot.
Response 8: we have deleted the title inside each plot to keep the manuscript uniform style.

Reviewer 3 Report
Authors presented an interesting and topical paper, which is linear organized, scientifically sound and well-motivated. Some interventions are required to further enhance manuscript merit. Authors should underline that security issue in IoT is more relevant wrt to traditional computer systems due to the cyberphysicality of IoT devices (they can perform actual and relevant actions on the physical world) and due to the sensitive nature of some data (privacy for medical data, personal routine data, etc.) (in these directions, refer to <Fortino, G., Rovella, A., Russo, W., & Savaglio, C. (2016). Towards cyberphysical digital libraries: integrating IoT smart objects into digital libraries. In Management of Cyber Physical Objects in the Future Internet of Things (pp. 135-156). Springer, Cham.>). Moreover, it should be specified which elements (e.g., IP, MAC address?) actually enables the IoT identification, to help readers who are not in this field. Perform a deep proof-reading in order to enahnce manuscript overall style (re-phrase some statements for the sake of readability e.g., "Section 2 explains the related work. Section 3 is developed to point out the modeling and processing of GAN", use past tense verbs for conclusion) and presentation (check if figures captions and subsections titles should start with lower or upper case letters, uniform reference style e.g., "company [3]" and "address[6]"), and to fix minors (Fig. 4_the feature distribution" --> "Fig.4_Features Distribution" and others, "The recognition rate" -> "Recognition Rate" on plot axis legend, "Moreover, An improvement") and typos ("maximinze","pervious"). Improve bibliography by adding a couple of references within introducion (e.g. a general reference for classification and machine learning algortihms between lines 34-50, add <Fortino, G., Russo, W., Savaglio, C., Viroli, M., & Zhou, M. (2017, June). Modeling opportunistic iot services in open iot ecosystems. In 17th Workshop From Objects to Agents WOA(pp. 90-95).> at line 19 as genera reference for IoT applications).
In conclusion, paper has merit but the aforementioned issues should be addressed before its full acceptance
Author Response
"Fig.4_Features Distribution" and others, "The recognition rate" -> "Recognition Rate" on plot axis legend, "Moreover, An improvement") and typos ("maximinze","pervious").
Response 3: We have checked the manuscript and revised it according to the comments.
First of all, we have rephrase "Section 2 explains the related work. Section 3 is developed to point out the modeling and processing of GAN" as “ Section 2 explains the use of physical layer security, and surveys the previous research related to OCC and GAN. Section 3 is developed to point out the structure of the basic GAN model and provides the framework of the generator and the discriminator model.”
Then, we carefully revised the conculsion section, where the past tense verbs are used foe conclusion. The experssions are shown as “Based on the open-categorical risk theory, this paper proposed the OCC-GAN model, which generated the fake samples to be evenly distributed in the open space. Moreover, an improvement algorithm of incremental OCC-GAN enabled the model to continuously integrate new sample classes with a high recognition rate of OCC. The final identification model can automatically extract features and have a strong robustness. Even if it dealt with the micro-data, the model can still achieve good recognition results. We conducted a series of experiments to verify the performance of our proposed model. Comparing to other common methods, such as OC-SVM and OvR-SVM, the OCC-GAN model had a better result . This model can achieve an accuracy rate of more than 90% with a recall rate of 97%.”
Last, we are very sorry for our incorrect writing of the presentation that the reviewer provide. In this time, we have check the manuscript carefully to avoid all detail errors.
Point 4: Improve bibliography by adding a couple of references within introducion (e.g. a general reference for classification and machine learning algortihms between lines 34-50, add
Response 4: It is really true as reviewer suggested that we should improve bibliography by a couple of references within introduction. Here, we learn the reference that the reviewer provided, and lin 19 “such as indoor electronic alarm, temperature alarm, traffic signal and so on. ” were corrected as “ e.g., a smart city in which the IoT device belonging to different application contexts cooperate for providing services related to e-health, smart factories, energy and traffic management [1,2]”
[1] Fortino, G.; Russo, W.; Savaglio, C.; Viroli, M.; Zhou, M. Modeling opportunistic iot services in open iot
ecosystems. 17th Workshop From Objects to Agents WOA, Italy, Jun. 2017, pp. 90–95.
[2] Molina, B.; Palau, C.E.; Fortino, G.; Guerrieri, A.; Savaglio, C.
Empowering smart cities through interoperable Sensor Network Enablers. Systems, Man and Cybernetics (SMC), 2014 IEEE International Conference on, 2014, pp. 7–12.

Round 2
Reviewer 1 Report
The authors have answered all my questions and I am quite satisfied with their responses.
Hence, I want to accept the paper in present form.